# Multi-Level Analysis of Adipose Tissue Reveals the Relevance of Perivascular Subpopulations and an Increased Endothelial Permeability in Early-Stage Lipedema

**DOI:** 10.3390/biomedicines10051163

**Published:** 2022-05-18

**Authors:** Karin Strohmeier, Martina Hofmann, Jaroslaw Jacak, Marie-Sophie Narzt, Marlene Wahlmueller, Mario Mairhofer, Barbara Schaedl, Wolfgang Holnthoner, Martin Barsch, Matthias Sandhofer, Susanne Wolbank, Eleni Priglinger

**Affiliations:** 1Ludwig Boltzmann Institute for Traumatology in Cooperation with the AUVA, 1200 Vienna, Austria; karin.strohmeier@trauma.lbg.ac.at (K.S.); marie.narzt@trauma.lbg.ac.at (M.-S.N.); marlene.wahlmueller@trauma.lbg.ac.at (M.W.); barbara.schaedl@trauma.lbg.ac.at (B.S.); wolfgang.holnthoner@trauma.lbg.ac.at (W.H.); susanne.wolbank@trauma.lbg.ac.at (S.W.); 2Austrian Cluster for Tissue Regeneration, 1200 Vienna, Austria; martina.hofmann@fh-linz.at (M.H.); jaroslaw.jacak@fh-linz.at (J.J.); 3School of Medical Engineering and Applied Social Science, University of Applied Sciences Upper Austria, 4020 Linz, Austria; 4Department of Hematology and Internal Oncology, Johannes Kepler University, 4020 Linz, Austria; mario.mairhofer@jku.at; 5University Clinic of Dentistry, Medical University of Vienna, 1090 Vienna, Austria; 6Austrian Center for Lipedema, Skin Aesthetic Vein Laser Practice, 4020 Linz, Austria; martin@drbarsch.at (M.B.); dr.matthias@sandhofer.at (M.S.)

**Keywords:** lipedema, endothelial permeability, endothelial cells, pericytes, stromal vascular fraction (SVF), adipose-derived stromal/stem cells (ASC), adipose tissue, aromatase, ZNF423, machine learning

## Abstract

Lipedema is a chronic, progressive disease of adipose tissue with unknown etiology. Based on the relevance of the stromal vascular fraction (SVF) cell population in lipedema, we performed a thorough characterization of subcutaneous adipose tissue, SVF isolated thereof and the sorted populations of endothelial cells (EC), pericytes and cultured adipose-derived stromal/stem cells (ASC) of early-stage lipedema patients. We employed histological and gene expression analysis and investigated the endothelial barrier by immunofluorescence and analysis of endothelial permeability in vitro. Although there were no significant differences in histological stainings, we found altered gene expression of factors relevant for local estrogen metabolism (aromatase), preadipocyte commitment (ZNF423) and immune cell infiltration (CD11c) in lipedema on the tissue level, as well as in distinct cellular subpopulations. Machine learning analysis of immunofluorescence images of CD31 and ZO-1 revealed a morphological difference in the cellular junctions of EC cultures derived from healthy and lipedema individuals. Furthermore, the secretome of lipedema-derived SVF cells was sufficient to significantly increase leakiness of healthy human primary EC, which was also reflected by decreased mRNA expression of VE-cadherin. Here, we showed for the first time that the secretome of SVF cells creates an environment that triggers endothelial barrier dysfunction in early-stage lipedema. Moreover, since alterations in gene expression were detected on the cellular and/or tissue level, the choice of sample material is of high importance in elucidating this complex disease.

## 1. Introduction

Lipedema was originally described by Dr. Allen and Dr. Hines in the 1940s as a clinical syndrome that is characterized by an increase in subcutaneous adipose tissue in the buttocks and lower extremities, which is associated with pain and limited mobility [1]. To date, only liposuction under tumescence anesthesia helps to reduce pain, pressure, and the tendency to hematoma [2,3,4,5,6,7]. It has been reported that 89% of lipedema patients have maternal and paternal predisposition over three generations [8,9]. The epidemiological prevalence of lipedema is not clearly described but most experts are convinced that lipedema is a common, often overlooked clinical condition, affecting mostly women, and develops during phases of hormonal changes [10]. A total of 64% of patients reported the onset of symptoms at the end of puberty and the remaining patients saw the first signs of illness after pregnancy or menopause [11]. Therefore, estrogen has been suggested as the key regulator of the excessive adipose tissue accumulation in lipedema. Since estrogen directly affects lipid metabolism in white adipose tissue, mainly through its estrogen receptors (ER) alpha and beta, it has been hypothesized that altered ER expression and signaling might be involved in lipedema onset [12].

Different studies have postulated a correlation of the onset and progression of lipedema with microangiopathy, lymphangiopathy, adipocyte hyperplasia/hypertrophy, tissue hypoxia, fibrosis, and macrophage infiltration, without the causal triggering factors being found or the pathophysiological significance of the hormones being clarified [13,14,15,16,17,18,19]. Microangiopathy associated with the fat growth very early in the disease might lead to a disruption of the endothelial barrier function and to an increase in the permeability of the capillaries [14,20]. As a result, protein-rich fluid from the vascular system could enter the tissue [21]. In addition, macroangiopathy also develops, which mainly affects the venous vasculature. In consequence, an overall increase in the hydrostatic pressure in the venous system can be observed, which contributes to the formation of edema [22]. In the early stages of the disease, the lymphatic system is not influenced, and the lymph drainage works efficiently [14]. As the disease progresses, overstretched, dilated lymph vessels [17] have been reported with higher permeability for fluid and macromolecules [23]. The chronic fluid exposure of the tissue can cause secondary inflammatory reactions, increasing fibrosis [24]. Recently, intercellular fibrosis and adipocyte hypertrophy have been observed, alongside an unaltered morphology of the lymphatic vessels. Despite this, lymphatic-related cytokine secretion could be identified, which might relate to vascular permeability [18,19]. The most recent research demonstrated an enhanced oxidative metabolic capacity of the stromal vascular fraction (SVF) in lipedema patients [25]. Lipidomics, metabolomics, and transcriptional profiling of adipose tissue, adipocytes and adipose-derived stromal/stem cells (ASC) exhibited significant differences between lipedema and healthy individuals [26]. We have previously shown that the SVF cells constitute an important player in lipedema disease. In SVF-derived extracellular vehicles (EVs), we identified a characteristic profile of microRNAs (miRNAs), with known impacts on cellular processes affected by lipedema, such as adipogenesis, angiogenesis, inflammation, and fat metabolism [27]. Furthermore, we observed a reduction of in vitro adipogenic differentiation of ASC and an enhanced cell number of mesenchymal/pericytic subpopulations in lipedema [28].

Recognizing the relevance of stromal vascular subpopulations in local events in the diseased tissue, we aimed at identifying more closely the different cell types involved in lipedema onset and progression, specifically focusing on cells interacting with the local vascular endothelium. From lipoaspirates as a starting material, we obtained the SVF and isolated endothelial and pericyte subpopulations, as well as ASC cultures. We applied a thorough analysis, including an in vitro setup, to study the vascular endothelial barrier function on the adipose tissue itself, down to the cellular level.

## 2. Materials and Methods

### 2.1. Study Participants and Experimental Set-Up

The collection of human adipose tissue was approved by the local ethical board with written patient consent. Subcutaneous adipose tissue (AT) was obtained from the lateral thighs during routine outpatient liposuction procedures under local tumescence anaesthesia. The harvesting triport cannulas were 4 mm in diameter (MicroAire System power-assisted liposuction). As an endogenous control, AT of the abdominal region of lipedema patients, which is not affected by lipedema, was included into the study. Female non-obese healthy subjects (ctrl, *n* = 4) at the age of 37.8 ± 2.2 with a BMI of 23.3 ± 3.0 and lipedema patients (lip, *n* = 5) at the age of 33.6 ± 10.7 with a BMI of 29.5 ± 2.5 were included into this study. Lipedema was defined by Dr. Sandhofer and Dr. Barsch (Center for Lipedema, Austria) via clinical anamnesis, including inspection (disproportional enlargement of specific subcutaneous fat (knee, ankle, no swollen forefoot, and toes)), ultrasound (phlebological analysis, typical ‘snowstorm’ pattern and absence of fluid accumulation) and impedance measurement (fat analysis). Lipedema stage 1–2/type I-III according to Sandhofer et al. [3] were included into the study. A schematic representation of the experimental set-up is shown in Figure 1.

### 2.2. Histology

Biopsies from ctrl and lip individuals were taken from the dermal and subcutaneous thigh region and fixed with 4% neutral buffered formalin (Roti, Histofix, Carl Roth, Karlsruhe, Germany) for 24 h and rinsed in phosphate-buffered saline (PBS). The samples were then dehydrated in a graded series of ethanol and embedded in paraffin via the intermedium Xylol. Sections of 4 µm were performed on a rotary microtome (HM355S Microm; Thermo Fisher Scientific, Waltham, MA, USA) and then deparaffinized and rehydrated for either histochemical or immunohistochemical analysis. The sections were histochemically stained with haematoxylin and eosin (H/E) for overview and Martius Scarlet blue (MSB) staining was performed for detecting fibrin and collagen. Immunohistochemical staining of the biopsies was carried out for cell identification using markers indicative of endothelial cells (CD31), pericytes (smooth muscle actin, SMA), macrophages (CD68) or lymphatic endothelial cells (Podoplanin). The antigen retrieval was performed by either steaming the sections for 20 min in a Tris-EDTA buffer (pH 9) for SMA or sodium citrate buffer (pH 6) for CD31, or incubating the sections with Proteinase K (Dako, Santa Clara, CA, USA) for CD68. For the incubation with the anti-podoplanin antibody, no antigen retrieval was necessary. After retrieval, the sections were rinsed in TBS and further prepared by blocking the endogenous peroxidase and alkaline phosphatase with BLOXALL (Vector Laboratories, Burlingame, CA, USA). After another rinsing step with TBS, the sections were incubated with the antibodies for 1 h at room temperature. The concentrations of the antibodies were used as following: CD31 (DAKO, Vienna, Austria) 1:25, SMA (Merck, Darmstadt, Germany) 1:5000, CD68 (Thermo Fisher Scientific, Waltham, MA, USA) 1:100 and Podoplanin (Biorad, Hercules, CA, USA) 1:40. The secondary antibody Brightvision poly HRP (Immunologic, Duiven, The Netherlands) was incubated for 30 min. Visualization was performed with the peroxidase substrate NovaRED™ (Vector Laboratories, Burlingame, CA, USA) and counterstaining was performed with hematoxylin. Images were taken on a light microscope (Nikon E800, Vienna, Austria) and analyzed using QuPath-0.2.3 (University of Edinburgh).

### 2.3. SVF Isolation

A total of 200 mL of liposuction material was washed with an equal volume of PBS. Tissue digestion was carried out with 0.2 U/mL collagenase NB4 (Nordmark, Uetersen, Germany) dissolved in 200 mL of PBS, containing Ca^2+^/Mg^2+^ and 25 mM of N-2-hydroxyethylpiperazine-N0-2-ethanesulfonic acid (HEPES; Sigma-Aldrich, St. Louis, MO, USA) at 37 °C under moderate shaking (180 rpm) for 1 h, resulting in an end concentration of collagenase of 0.1 U/mL. The digested tissue was transferred into 50 mL tubes (Greiner, Kremsmünster, Austria). After centrifugation at 1200× *g* for 7 min, the cell pellets were incubated with 50 mL of erythrocyte lysis buffer (154 mM of ammonium chloride (Sigma-Aldrich), 10 mM of potassium bicarbonate (Sigma-Aldrich), 0.1 mM of ethylenediamine-tetraacetic acid (EDTA; Biochrom, Austria) in Aqua dest for 2 min at room temperature. After centrifugation for 5 min at 500× *g*, the cell pellet was washed with PBS and filtered through a 100-µm cell strainer (Greiner, Austria). After another centrifugation step at 500× *g* for 5 min, the supernatant was removed, and cell number and viability were determined using the NucleoCounter^®^ NC-200™ (Chemometec, Allerod, Denmark). To allow accurate quantification of the freshly isolated SVF cells, the protocol “Viability and Cell Count—Aggregated Cells Assay” was performed. ASCs were analyzed using the protocol “Viability and Cell Count Assay” according to the manufacturer’s instructions.

### 2.4. Isolation of Endothelial Cells and Pericytes by Fluorescence-Activated Cell Sorting (FACS)

Freshly isolated SVF cells were resuspended in EGM-2 medium, filtered through a 40 µm cell strainer (Greiner, Austria), and cell yield was determined. The 5 × 10^6^ cells were incubated with 25 µL of primary antibodies anti-CD31-FITC (BD Biosciences, Austria), anti-CD146-PerCP (R&D, Germany) and anti-CD45-PE (BD), for 30 min at 4 °C in 250 µL of FACS Buffer (PBS, 0.1% BSA, 0.2 mM Glutamine) or remained unstained to serve as the control. Afterwards, the cells were washed with FACS Buffer, centrifuged at 300× *g* for 5 min, resuspended in 2.5 mL of FACS Buffer and filtered through a 40 µm cell strainer. Cell sorting was performed on a FACS Aria I (Becton Dickinson, Heidelberg, Germany) using a 100 µm nozzle to yield two distinct cell populations, which were as follows: endothelial cells sorted as CD31+, further referred to as ‘EC’ and pericytes sorted as CD45−/CD31−/CD146+, further referred to as ‘PC’. Unstained cells were used to determine the background autofluorescence. Cell aggregates were excluded in an FSC-W versus FSC-A gate. Sorting was performed in ‘Purity’ mode and approximately 3 × 10^5^ EC and PC were collected in 1 mL of EGM-2. After centrifugation at 300× *g* for 5 min, the cell pellets were resuspended in 800 µL of TRI Reagent^®^ (Sigma-Aldrich). Until RNA isolation, the samples were stored at −80 °C.

### 2.5. ASC Cultivation

Freshly isolated SVF cells were seeded at a density of 1 × 10^6^ cells per T-25 culture flask and cultured in EGM-2 at 37 °C, 5% CO_2_, and 95% air humidity. The medium was changed every 3 to 4 days. When cells reached a subconfluent state, they were passaged, and cell number and viability were determined. ASCs were seeded at a density of 1 × 10^5^ in T-25 culture flasks and cultured until passage two (ASC P2).

### 2.6. Cultivation of Human Primary EC

Human umbilical cord blood-derived CD34+ cells differentiated to EC (hEC) were provided in frozen aliquots of 10^6^ cells at passage five by Prof. Fabien Gosselet (Université d’Artois, France). The hECs were seeded at a density of 1 × 10^6^ cells per 10-cm cell culture dish (Corning, NY, USA), precoated with 0.2% pig skin gelatine type A (Sigma-Aldrich) and cultured in ECM-5 for 2 days before the conduction of experiments.

### 2.7. Immunofluorescence Staining of Endothelial Junctions

To compare the intrinsic junctional organization of EC derived from healthy and lipedema individuals, the freshly isolated SVF cells (ctrl-SVF and lip-SVF, respectively) were seeded at a density of 3 × 10^4^ in 8-well chambered coverglass (Thermo Fisher) in EGM-2 and cultured for 6 days.

For imaging alterations in endothelial junctions induced by secreted factors from the SVF, 6 × 10^6^ SVF cells were seeded in a T-175 cell culture flask with 22 mL of Endothelial Cell Medium (ECM)-5 (ECM basal medium (Sciencell, Carlsbad, CA, USA) supplemented with 5% heat-inactivated fetal calf serum, 1% EC growth supplement (Sciencell), and 0.5% gentamycin). After 24 h, the conditioned medium (CM) was collected, centrifuged at 500× *g* for 5 min and the supernatant was stored at −80 °C until use. The hECs were seeded at a density of 5 × 10^4^ in glass bottom 8-well chamber slides pre-coated with 0.2% gelatine (Sigma-Aldrich). After 2 and 5 days, the medium was replaced with CM of SVF cells isolated from healthy subjects (ctrl-CM) and lipedema patients (lip-CM). After 6 days of cultivation, the cells were washed with PBS (containing Ca^2+^/Mg^2+^) and fixed with 1% PFA (paraformaldehyde) for 30 min. The fixed cells were permeabilized with 0.1% Triton X-100 (Sigma-Aldrich) for 10 min and blocked in 1% albumin from chicken eggs (ACE; Sigma-Aldrich) for 1 h. The cells were then labelled with anti-CD31 AlexaFluor 488 and anti-ZO-1-1A12 AlexaFluor 647 (Thermo Fisher) diluted in 1% ACE (dilution 1:200) for 1 h in the dark. The cells were washed with PBS and images were taken on a modified custom-built Olympus IX81 inverted epi-fluorescence microscope with an oil-immersion objective (PlanApo N 60× 1.42 NA, Olympus). A tube lens with an additional magnification of 1.6 was used, resulting in a final magnification of 96x. The sample was positioned on a mechanical stage (travel range of 1 cm × 1 cm, Hybrid, JPK Instruments) with nanometer precision XYZ stage. Images were detected with an iXonEm+ 897 (back-illuminated) EMCCD camera (16 µm camera pixel size; Andor, Oxford Instruments, Belfast, UK). The samples were illuminated using a solid-state laser with a wavelength of 640 nm (diode-pumped, Toptica Photonics, Graefelfing, Germany) and a 487 nm solid-state laser (diode-pumped, Toptica Photonics). The following filter sets were used: dichroic filter (ZT405/488/561/640rpc, Chroma, Olching, Germany) and emission filters: ET 700/75 M (Chroma Technology GmbH) and ET 525/50 M (Chroma Technology GmbH, Olching, Germany). The laser illumination protocols, the camera and the stage were controlled by a custom-written software called QStdControl (C++, Qt framework) [29,30]. The images were processed with Fiji (ImageJ 1.53c).

### 2.8. Convolutional Neural Network (CNN) Analysis of Endothelial Junction Imaging

We used machine learning of artificial neural networks for classification of the fluorescence microscopy images. A detailed description and additional information to our applied method is presented in the Appendix A. In short, we applied the deep learning method convolutional neural networks (CNN). The CNN consisted of an input layer, grouped hidden layers and an output layer. Each grouped CNN hidden layer included a convolution layer, a rectified linear unit (ReLU) layer, a pooling layer, and optionally a normalization layer. The output layers formed a fully connected layer and a classification layer. The CNN input data were an image matrix with the size 250 × 250 pixels for each individual red, green, and blue channel. Each image was recorded in two channels, ZO-1 in the red channel (excitation at 647 nm) and CD31 in the blue channel (excitation at 492 nm). To increase the dataset, the image-data store was augmented with transformations of original images and combined images were saved in the green channel. After the augmentation and transformation process, each original image was represented by a group of 9 RGB images. For the first set of data (SVF-EC), we used 25 original images of lip-SVF and 33 original images of ctrl-SVF. To overcome the imbalance of the number of samples in the two groups, we generated duplicates of the underrepresented group for the training set. A total of 5 original images of each group were randomly selected as the test data set, resulting in 45 images per group/class (9 augmented images per original image). The remaining 252 images per group formed the training set. For the second set of data (CM-treated hEC), 120 original ctrl-CM images and 90 original lip-CM images were available. The groups were balanced by generating duplicates of images to train the network. A total of 9 RGB images were generated for each original image, which gave a total of 1080 RGB images per group, 936 images per group were included in the training data set and 144 images per group formed the test data set. After performing the training and test sessions, we calculated the recognition/classification accuracy of the network. The trained CNN was then used to calculate the posteriori probability of classification of individual images to the lip and ctrl group.

### 2.9. Endothelial Permeability Assay and RNA Collection of hECs

To determine endothelial permeability, a transwell assay was performed. Transwell 0.4 µm pore inserts (Corning) for 24-well plates (growth area 0.33 cm^2^) were coated with 150 µL of Matrigel (BD Biosciences, Franklin Lakes, USA) at a concentration of 0.19 mg/mL dissolved in DMEM-LG (Lonza, Switzerland) for one hour at room temperature. Then, the inserts were washed with DMEM-LG and equilibrated with 100 µL of ECM-5. 2.5 × 10^5^. The hECs were seeded per insert and cultured in 100 µL of ECM-5; 600 µL of ECM-5 was pipetted in the lower compartment. After 2 and 5 days, the medium of the inserts and the lower compartment were replaced with ctrl-CM and lip-CM. The assay was performed in triplicates for each donor and three filters were treated with ECM-5 only as a control. For calculation, three filters without cells were included in the assay. After 6 days of cultivation in the transwell inserts, endothelial permeability assay was performed. As a washing step, the transwell inserts were placed in fresh 24-well plates containing warm Ringer-HEPES/0.1% BSA. The medium of the filters was carefully replaced with 1 µM of sodium-fluorescein solution (Na-F) and the transwell inserts were immediately transferred to fresh wells containing warm Ringer-HEPES/0.1% BSA. The plates were incubated for 20 min in a shaking incubator at 37 °C and 60 rpm, followed by another 2 transfers to fresh wells with warm Ringer-HEPES/0.1% BSA, resulting in the following three time points: 0–20 min, 20–40 min, and 40–60 min. Permeability of the EC layer was determined by measuring the Na-F concentration in the lower and upper (transwell inserts) compartments. The samples of all the compartments and Na-F standard solutions (2–1000 nM) were transferred to clear bottom, black-walled 96-well plates (Greiner) and fluorescence intensity (excitation 480 nm, emission 525 nm) was measured on a Tecan infinite 200 pro plate reader (Tecan, Austria). Endothelial permeability (Pe (×10^−3^ cm/min)) was calculated as described elsewhere [31].

For gene expression analysis of selected endothelial junction genes, hECs were cultured in parallel in 24-well plates and were likewise treated with ctrl-CM and lip-CM from day 2 to day 6. On day 6, the cells were trypsinized and collected in TRI Reagent (Sigma-Aldrich) for RNA isolation. For immunofluorescence imaging of endothelial junctions, transwell membranes were cut out from the inserts using a scalpel, labelled with anti-ZO-1-1A12 AlexaFluor 647 antibody (Thermo Fisher) as described above and sandwiched between two coverslips. Imaging was performed as described above, except for the utilization of a 20× magnification objective (Olympus).

### 2.10. RNA Isolation and Quantitative Real-Time RT-PCR

A total of 200 µL of PBS-washed adipose tissue were transferred to a 1.5 mL tube and 800 µL of TRI Reagent were added. The samples were vigorously vortexed and homogenized in a Thermomixer (Comfort, Eppendorf) for 2 h at 15,700× *g* at room temperature. The total RNA of AT (4 replicates per donor), SVF cells (2 replicates per donor), as well as SVF-derived sorted EC/PCs and hECs was isolated using TRI Reagent (Sigma-Aldrich), according to the manufacturer’s protocol. Isolated RNA concentration and purity were measured on a Biophotometer 6136 (Eppendorf). cDNA was transcribed from 500 ng RNA per sample using the HighCapacity cDNA Reverse Transcription Kit (Applied Biosystems, Vienna, Austria). Quantitative real-time RT-PCR (qPCR) was conducted in technical duplicates using the TaqMan Fast Advanced Master Mix (Applied Biosystems) and TaqMan gene expression assays (Applied Biosystems). The primers to *CYP19A1* (Hs00903411_m1), *ESR1* (Hs01046816_m1), *ESR2* (Hs01100353_m1), *PPARG* (Hs01115513_m1), *LEP* (Hs00174877_m1), *FABP4* (Hs01086177_m1), *ADIPOQ* (Hs00605917_m1), *ZNF423* (Hs00323880_m1), *ITGAX* (Hs00174217_m1), *TEK* (Hs00945142_m1), *SELE* (Hs00174057_m1), *CDH5* (Hs00901469_m1) and *TJP1* (Hs01551871_m1) were normalized to *UBC* (Hs00824723_m1) as housekeeping genes. PCR and cDNA synthesis was performed on a CFX96 Real-Time PCR Detection System (Bio-Rad Laboratories, Vienna, Austria). Relative gene expression of target genes was calculated using the 2^−∆∆Ct^ method.

### 2.11. Protein Array Raybio C-Series

To screen the secretome of SVF cells isolated from the thigh region of healthy subjects and lipedema patients, 5 × 10^5^ freshly isolated SVF cells were seeded in a T-25 cell culture flask with 2.5 mL EGM-2 medium, which was replaced with serum-free DMEM low glucose with 2 mM of L-glutamin (Lonza) after 2 h. After 24 h, the CM was collected and stored at −80 °C until further use. The CM was subjected to a customized membrane-based sandwich immunoassay (RayBio^®^ Membrane-Based Antibody Arrays (C-Series)) according to the manufacturer’s instructions (RayBiotech, Peachtree Corners, GA, USA). Imaging of the membranes was performed on a Vilber Lourmat Fusion-SL-3500 WL (Vilber, France) and analyzed using FusionCapt Software (Vilber). The images were then analyzed with the Fiji Plugin “Protein Array Analyzer” (ImageJ) and normalized according to the manufacturer’s provided analysis table.

### 2.12. Statistics

All statistical analyses were performed using Graphpad Prism 8.01. The values were expressed as the mean ± standard deviation (SD). A non-parametric unpaired Mann–Whitney U-test was performed for non-Gaussian distribution, whereas a two-tailed Student’s *t*-test was performed for Gaussian distribution. In case of unequal variances, a Welch’s correction was performed. For multiple group comparison, a one-way ANOVA with Tukey’s post-hoc test was used. Statistical significance was set at *p* < 0.05 (* *p* < 0.05, and ** *p* < 0.01).

## 3. Results

### 3.1. Skin Biopsies of Lipedema Patients Showed No Morphological Alterations

Dermal biopsies from healthy and lipedema individuals were histochemically or immunohistochemically stained and quantitatively characterized. No difference in epidermis thickness was detected in the H/E staining (Figure 2a). MSB staining showed no signs of fibrosis in lipedema patients compared to healthy individuals (Figure 2b). The immunohistological analysis of the endothelial marker CD31 (Figure 2c) showed a slight, but not significant, difference between both groups, whereas the von Willebrand factor (data not shown), the pericytic marker SMA (Figure 2d), and the lymphatic endothelial marker podoplanin (Figure 2e) showed no difference between the ctrl and lip samples. Interestingly, in areas of subdermal blood vessel networks and identified as CD31 positive and podoplanin negative, a higher number of macrophages (CD68+) is visible in lipedema skin, which may indicate blood vessels with altered permeability (Figure 2f). However, when quantification across the whole tissue section is conducted, including the affected subcutaneous adipose tissue (data not shown), this difference in visible macrophage infiltration could not be confirmed.

### 3.2. Gene Expression Analysis Revealed a Role of Local Estrogen Metabolism, Preadipocyte Determination and Immune Cell Infiltration in Lipedema

We next aimed at identifying cell types of the subcutaneous adipose tissue involved in lipedema pathology. Therefore, we isolated the SVF of liposuction material from the thigh region of both healthy and lipedema individuals (ctrl and lip thigh, respectively) and included liposuction material of the abdominal region of lipedema patients (lip abdomen) in all the analyses. Since the abdominal AT site is described as not being affected by lipedema, it serves as an endogenous control for the lipedema-affected thigh AT. From all the SVF samples, CD31+ EC and CD45− CD31− CD146+ PC were isolated by FACS (Figure 3a–e shows the gating strategy) and ASCs were collected after two passages of SVF cells. There was no difference in relative PC and EC counts between the control and lipedema samples (Figure 3f).

Gene expression analysis employing qPCR was performed on liposuction material directly (AT), the SVF, EC, PC, as well as cultured ASC P2 (Figure 4).

#### 3.2.1. Estrogen-Related Genes

To determine site-specific and cell-specific expression differences of genes involved in estrogen metabolism and signaling, we analyzed gene expression of aromatase (*CYP19A1*), the enzyme responsible for converting androgens to estrogens, and estrogen signaling receptors ER alpha (ERα, gene name *ESR1*) and ER beta (ERβ, gene name *ESR2*). *CYP19A1* shows a significant upregulation in lipedema thigh AT compared to control thigh AT (*p* = 0.048). Of note, *CYP19A1* mRNA levels are significantly upregulated in lipedema thigh, compared to lipedema abdominal AT (*p* = 0.022) of the same individuals. There is a tendency of an increased aromatase expression in isolated SVF in lipedema thighs, compared to control thighs (*p* = 0.094) and lipedema abdominal SVF (*p* = 0.098), both without significance due to high donor variability. For both estrogen receptors (*ESR1*, *ESR2*), no significant difference was observed between healthy and diseased tissue or cells thereof. mRNA expression of the estrogen receptor, *ESR1*, shows a tendency of upregulation in lipedema compared to healthy thigh-SVF, although without significance (*p* = 0.075) (Figure 4). Interestingly, cultured ASC P2 showed the highest *CYP19A1* expression compared to the other samples analyzed, although without being differentially expressed between control and lipedema thighs (Appendix A).

#### 3.2.2. Adipogenic Marker Genes

As markers of adipogenic lineage determination and adipocyte differentiation, we evaluated the gene expression of *ZNF423*, *PPARG*, *FABP4*, *LEP* and *ADIPOQ*. One gene that controls early preadipocyte determination, *ZNF423* [32], showed a significant upregulation in lip-thigh-SVF (*p* = 0.037) and lip-thigh-PC (*p* = 0.045) compared to the respective control groups. Importantly, this difference was even more pronounced in sorted lip-thigh-EC (*p* = 0.002). Surprisingly, lipedema-derived abdominal EC also showed a significantly increased *ZNF423* gene expression (*p* = 0.019). A key transcription factor in adipocyte differentiation, peroxisome proliferator activated receptor-γ (*PPARG*), showed a slight trend of increase in PC derived from lipedema thigh tissue compared to healthy thigh tissue (*p* = 0.086). The gene expression of fatty acid binding protein 4 (*FABP4*), which is induced during adipocyte differentiation [33], showed no significant difference between any of the groups. Leptin (*LEP*) and adiponectin (*ADIPOQ*), markers mainly expressed by mature adipocytes, showed no significantly different gene expression in any sample types, suggesting an alteration of early preadipocyte determination, rather than late differentiation in lipedema (Figure 4). Of note, cultured unstimulated ASC P2 showed the lowest expression of adipogenic markers compared to the other samples (Appendix A).

#### 3.2.3. Immune Cell Marker Gene

To confirm that previously postulated inflammatory processes are involved in the affected tissue [19], we selected CD11c (*ITGAX*) as a marker gene for immune cell infiltration. Indeed, *ITGAX* showed a significantly increased gene expression in AT of lipedema thighs (*p* = 0.022) compared to control thighs, whereas the isolated SVF cells showed no significant difference in any group (Figure 4). In addition, EC, PC and ASC show *ITGAX* expression, although mRNA levels throughout all the samples were relatively low (Appendix A).

In summary, our gene expression analysis revealed a possible involvement of local estrogen metabolism, as well as a dysregulation of *ZNF423* in SVF, EC and PC, and hence a possible role of endothelium-related cells (EC and PC) in lipedema. Of note, there were no differences in gene expression detected in cultivated, untreated SVF-derived ASC P2 of either control or lipedema subjects in any of the selected genes.

### 3.3. Lipedema Is Characterized by Altered Endothelial Cell Junctions and Higher Endothelial Permeability

It has been postulated that lipedema is accompanied by leaky vessels, leading to an increased passage of protein-rich fluid into the affected tissue [21]. Hence, we wanted to determine if the cellular junctions of EC and the impact of SVF secreted factors on EC permeability are altered in lipedema.

#### 3.3.1. Endothelial Junction Morphology

We investigated the junctional morphology of EC present in the SVF culture (Figure 5b), as well as the impact of SVF secretome on a model of healthy human EC differentiated from umbilical cord blood (Figure 5c). To image endothelial junctions of SVF-derived EC, freshly isolated SVF cells of control and lipedema subjects were seeded and cultured for 6 days, to allow formation of intercellular tight junctions in EC ‘islets’. We performed immunofluorescence staining with the endothelial marker CD31, and the tight junction protein ZO-1 and employed machine learning of artificial neural networks (CNN) for unbiased analysis of morphological differences in endothelial junctions (Figure 5a). By calculating the posteriori probability of image classification, the trained CNN network identified if individual fluorescence images of EC junctions could be classified to the ctrl or lip group and were, thus, distinguishable. A total of 29 of 33 ctrl-SVF and 23 of 25 lip-SVF images were correctly recognized by the trained CNN, which resulted in an average classification accuracy of 89.7%. Only 4 images of the ctrl-SVF and 2 images of lip-SVF groups were falsely classified (10.3%) (Figure 5d). Hence, cell-cell junctions of ECs present in lipedema SVF can be distinguished from those of healthy subjects, demonstrating a different junctional morphology.

To investigate if secreted factors of SVF are able to induce a difference in endothelial junction morphology, we collected CM of control and lipedema SVF cells after 24 h of cell culture. As a model of healthy/functional EC junctions, human primary ECs (hEC) incubated with the collected CM of control and lipedema SVF cells and immunofluorescence experiments were conducted after 6 days (Figure 5c). A total of 116 of 120 ctrl-CM and 85 of 90 lip-CM images were correctly recognized by the CNN, which resulted in an average classification accuracy of 95.7%. Only 4 images of ctrl-CM and 5 images of lip-CM were falsely classified (4.3%) (Figure 5e). Hence, endothelial junctions of lip-CM-treated hECs can be distinguished from the healthy control group by the applied machine learning method, signifying differences in endothelial junction morphology induced by lipedema and healthy SVF secretome.

In summary, the EC contained in the SVF of lipedema patients and the milieu created by lipedema SVF cells impact endothelial cellular junctions. However, hECs treated with the secretome of SVF cells show a more distinguishable alteration between lipedema and healthy samples than the endothelial junctions of the endogenous EC present in SVF.

#### 3.3.2. Endothelial Barrier Function

In order to determine if the differences in endothelial junction morphology, detected in the CNN analysis, are accompanied by a weakened endothelial barrier, we treated a monolayer of hECs in transwell inserts with SVF-CM of healthy and lipedema individuals. After 6 days, we measured the leakage of the fluorophore Na-F through the endothelium and calculated endothelial permeability. After treatment of hECs with CM of lipedema SVF, there was a significant increase (*p* = 0.0154) in endothelial permeability compared to the control group (Figure 6b). This suggests that the endothelial barrier function is negatively affected by the secretome of lipedema-derived SVF. Immunofluorescence images of the tight junction protein ZO-1 show endothelial junction morphologies of healthy and lipedema CM treated cells on transwell inserts (Figure 6d).

To investigate the factors involved in the weakened endothelial barrier of lip-CM-treated hECs, we analyzed the gene expression of VE-cadherin (*CDH5*), ZO-1 (*TJP1*), E-selectin (*SELE*) and Tie-2 (*TEK*) (Figure 6c). The treatment of hECs with lip-CM led to a strong significant decrease in adherens junction protein *CDH5* gene expression compared to ctrl-CM incubated cells (*p* = 0.0075). There was a trend of reduced *TJP1* expression, coding for ZO-1, a component of endothelial tight junctions, although without significance (*p* = 0.0719). The gene expression of *TEK* and *SELE* showed no difference between both groups. Together, these data suggest an involvement of both adherens and tight junctions in the impaired endothelial barrier detected in hECs treated with CM of lipedema-derived SVF.

### 3.4. IL-8 Protein Secretion Is Down-Regulated in CM of Lipedema SVF Cells

To identify the proteins present in the CM of lipedema-SVF cells that contribute to the altered endothelial barrier function, we performed a semi-quantitative membrane-based antibody array. Levels of 29 selected proteins known to play a role in endothelial function and inflammatory processes were analyzed in CM of thigh-derived SVF from healthy and lipedema subjects (Table 1). IL-8 protein levels significantly decreased in CM of lipedema-derived SVF (*p* = 0.049). In all the other analyzed proteins, there was no significantly different secretion between healthy individuals and lipedema patients, and a high variability between donors within each group.

## 4. Discussion

In the present study, we conducted a multi-level analysis of the AT, the isolated SVF, as well as perivascular subpopulations of lipedema patients. In order to identify factors involved in the onset of lipedema disease, we included non-obese patients diagnosed with lipedema stage 1–2 and non-obese healthy control individuals in our study.

The analysis of skin biopsies did not reveal histological differences between lipedema patients and control subjects. In H/E stainings of skin biopsies, we could not observe a difference in the thickness of the epidermis, contrary to Felmerer et al., who measured significant thicker epidermis in biopsies of lipedema patients [18], possibly owed to the difference in disease stages of study participants.

As previously observed by our group, the number of CD31+ cells [28] was not altered in lipedema tissue, which was confirmed by FACS of SVF subpopulations. While others reported significantly increased numbers of CD31+ vessels within the lipedema-affected AT and an increase in blood and lymphatic vessel area [17,34], Felmerer et al. reported that lymph and blood vessels were indistinguishable in number, size and percent coverage [19]. This is in line with our findings of unaltered counting of SMA, CD31 and podoplanin positive vessels in whole tissue sections. Moreover, secretion of angiogenic proteins by SVF cells was also not affected by lipedema, suggesting that local angiogenesis is not altered in early-stage lipedema adipose tissue. However, since SVF does not include mature adipocytes, a possible role of this cell type cannot be ruled out. In our FACS analysis, relative CD146+ cell counts were not different between lipedema and control samples. We had previously identified increased numbers of CD146+ cells within the SVF population of lipedema, but without excluding CD45+ CD146+ cells, which may indicate a relative enrichment of CD146+ lymphocytes within lipedema tissue [35]. Alternatively, this may point towards an increase in CD146+ numbers within the EC population [36]. Although we observed enhanced macrophage numbers in close proximity to blood vessels in some subdermal regions, there was no significant difference when analyzing the whole adipose tissue section, contradicting the often-described macrophage infiltration in tissue biopsies [17,18]. On the molecular level, expression of the *ITGAX* gene (CD11c) significantly increased in lipedema thigh AT. CD11c positive macrophages and dendritic cells were reported to show a pro-inflammatory phenotype and increased in number in obese AT [37]. In lean AT, dendritic cells were found to be the main subpopulation of CD11c+ cells [38]. Since we detected no clear indication of increased infiltration of macrophages in the tissue of non-obese lipedema stage 1–2, a role of dendritic cells should be considered in lipedema. Surprisingly, EC, PC and ASC also show *ITGAX* expression, although mRNA levels throughout all the samples were relatively low. Considering that FACS represents an enrichment rather than a clear separation of cell populations, the low *ITGAX* expression could originate from low numbers of the remaining immune cells. Of note, for separation of the EC subpopulation employed here, CD31 was used as the only marker, without discriminating for the negligible CD45+ CD31+ blood cells (6.4 ± 3.7%).

Previous studies reported adipocyte hypertrophy in lipedema hypodermal adipose tissue by the measurement of adipocyte size in histological stainings [17,18,25] and increased expression of typical adipogenic genes, such as PPARγ, adiponectin and leptin, in differentiated ASC cultures derived from the SVF of healthy and lipedema subjects [34]. We could neither confirm an increased adipocyte size in histology (data not shown), nor higher gene expression of late adipogenesis markers in primary uncultivated cell samples. The latter was in line with the recent findings of Felmerer et al. [18], who likewise found no differential expression of PPARγ, adiponectin and leptin in lipedema-affected adipose tissue. Importantly, we observed gene expression of a transcription factor critically involved in early adipocyte commitment to be affected by lipedema. *ZNF423* is an essential regulator of preadipocyte determination [32,39,40,41,42,43,44], since it is necessary and sufficient to activate PPARγ expression and adipocyte differentiation in culture [40] and is needed for the initial formation of adipose tissue in vivo [32,45]. We detected a significantly increased *ZNF423* expression in lipedema thigh compared to healthy thigh-derived SVF, isolated PC, and an even stronger increase in isolated CD31+ EC. This is in line with Gupta et al., who reported that *ZNF423* expressing cells are located at the adipose vasculature as a subset of both PC and capillary EC. The authors hypothesized that ZNF423+ EC may undergo an endothelial-mesenchymal transition, serving as a pool of committed preadipocytes [39]. Our data on isolated subpopulations suggest that EC and PC commitment to the adipocyte lineage might contribute to the adipose tissue expansion in early lipedema stages. This adds to the repeatedly observed changes in the early steps of adipogenic differentiation in lipedema [14,24,28,34,46]. However, the expression of PPARγ, a down-stream target of ZNF423, only slightly increased in lipedema thigh PC, compared to healthy controls. We hypothesize that the committed preadipocytes might fail to fully differentiate to mature adipocytes capable of efficiently storing triglycerides, which might be compensated by recruitment of further perivascular cells committing to the adipocyte lineage, leading to the expansion of fat mass. Of note, EC sorted from the supposedly unaffected abdominal subcutaneous AT of lipedema patients likewise showed increased *ZNF423* expression. In a recent study, Al-Ghadban et al. found that elevated gene expression of PPARγ in adipocytes differentiated from ASC of abdominal adipose tissue of lipedema patients [34], suggesting alterations of both fat depots in lipedema.

Interestingly, the *ZNF423* gene harbors four estrogen response element binding sites [47] and its expression is induced by estradiol, as previously shown in breast cancer cells [47,48]. Since the onset of lipedema is linked to female hormonal changes, a role of estrogen is presumed. To our knowledge, the demonstration of increased *ZNF423* expression in lipedema-derived SVF, EC and PC provides for the first time a potential link of estrogen to preadipocyte commitment in lipedema. Katzer et al. recently proposed that estrogen-mediated dysregulation of lower body fat accumulation in lipedema might be explained by a higher ERα/ERβ ratio or an increased production of steroidogenic enzymes, such as aromatase, leading to paracrine signaling and ERα activation among local adipocytes [12]. Indeed, we observed increased aromatase (*CYP19A1*) gene expression in lipedema thigh AT, which is also roughly reflected in the SVF, a cell population that does not contain mature adipocytes. Interestingly, cultured ASC showed the highest overall aromatase expression. Together, this might suggest mature adipocytes and/or ASC as the most relevant source of local estrogen. We did not find significantly different gene expression of estrogen receptors, neither on the tissue level nor within SVF. To further elucidate local estradiol signaling, confirmation of increased estrogen protein levels in lipedema-affected adipose tissue, the role of transcriptional co-regulators of estrogen receptors and the use of alternative estrogen receptors in specific cells of lipedema adipose tissue will be required.

One hypothesis about the pathophysiology of lipedema involves an increased blood capillary permeability due to a loss of elasticity of connective tissue and subsequent leakage of protein-rich fluid into the extracellular space [49,50]. In the current study, tight junction protein ZO-1 and adherens junction protein CD31 immunostaining revealed morphological differences in the endothelial junctions between lipedema- and control-derived SVF EC. Moreover, the cell culture supernatant of lipedema SVF was sufficient to change junctional morphology in healthy primary hECs. Our employed machine learning method correctly classified ctrl and lip images at a high level of accuracy, suggesting that both the EC itself and the environment created by surrounding cells are affected by lipedema. The alteration of endothelial junctions was more pronounced in the hECs treated with SVF secretome than in the endogenous SVF-EC. Changes in endothelial permeability were reported to be associated with the redistribution of cell surface proteins, stabilization of focal adhesion bonds and the activation of matrix metalloproteases [51]. Importantly, the observed morphological differences in our study were accompanied by a significant functional impairment of the endothelial barrier, probably due to a decrease in gene expression of the central components of endothelial adherens and tight junctions, cadherin 5 and ZO-1, respectively. *TEK* (Tie-2) and *SELE* (E-selectin) mRNA levels were not affected by lipedema SVF-CM, while Felmerer et al. [19] recently reported a decreased Tie-2 gene expression in lipedema AT. To our knowledge, this is the first evidence that lipedema, in fact, is characterized by an increased endothelial leakiness. Secreted factors of the lipedema-derived SVF were sufficient to increase endothelial permeability of healthy EC in an experimental setting lacking the surrounding connective tissue. These findings point out the importance of the SVF secretome in lipedema endothelial dysfunction independently of the mechanical effects and of the connective tissue. Unfortunately, this experiment could not be conducted with ECs isolated from the SVF directly. The overgrowth by residual ASC in the prolonged cultivation of ECs meant that a pure EC monolayer was not implementable.

The analysis of the SVF secretome at the protein level revealed a decrease in the pro-inflammatory chemokine IL-8 in lipedema. Bauer et al. reported an increased IL-8 concentration in CM of undifferentiated ASC from lipedema patients, which decreased upon adipogenic stimulation to the equally low levels of the control cells [46]. The significant reduction in IL-8 secreted by lipedema SVF may correlate with higher local estradiol levels, which we and others have implicated with lipedema formation [12]. In the context of breast cancer cell lines, IL-8 has been identified as a regulatory target of estrogen receptor α and inhibited IL-8 levels as a key factor for limited angiogenesis and tumor invasion [52]. Consistent with Felmerer et al. who found no difference in the systemic levels of IL-6 and leptin [17,18], we could not detect altered levels of these proteins in the local SVF secretome. In addition to proteins, intercellular signaling can be mediated by factors such as lipids, nucleic acid, including miRNAs, or EVs. miRNAs have clearly been identified as critical regulators of EC function/dysfunction and the associated vascular pathologies; hence, they may contribute to the observed changes in endothelial barrier function [53]. We recently reported a characteristic miRNA profile in SVF CM, seven miRNAs (miR–16-5p, miR-29a-3p, miR-24-3p, miR-454-p, miR–144-5p, miR-130a-3p, let-7c-5p) contained in small EVs and one (miR-188-5p) was protein-bound [27]. Amongst others, miRNA target analysis identified Notch and SMAD/transforming growth factor beta (TGFβ) signaling pathways, which regulate blood vessel branching and vascular sprouting [54,55]. miR-16 is an antiangiogenic miRNA, the inhibition of which acts proangiogenic on VEGF and bFGF signaling pathways [53]. Inhibition of miR-24-3p in circulating exosomes from diabetic foot ulcer patients restores angiogenesis in vitro and in vivo [56]. Estrogen receptor-regulated miRNAs have already been studied in the literature to identify prognostic/predictive markers for cancer, amongst which miR-16-5p and miR-29a-3p have been identified [57,58,59]. Therefore, local estradiol as a key regulator in lipedema may regulate local tissue cells, such as vascular EC, via miRNA signaling.

## 5. Conclusions

In the current study, we confirmed for the first time experimentally that lipedema is characterized by a weakened endothelial barrier. Moreover, we found a potential link of estrogen to preadipocyte commitment and subsequent fat mass expansion via the increased expression of aromatase in adipose tissue and ZNF423 in perivascular subpopulations in lipedema. Our data emphasize the importance of a refined analysis of the cellular components that form the adipose tissue. While alterations of aromatase and CD11c expression were only observed in AT, ZNF423 would have been overlooked without separation of the tissue into distinct cell types. Cultivated, untreated ASC showed the highest aromatase gene expression compared to the other samples, but the expression of estrogen receptors and adipogenic markers (except PPARγ) was the lowest in ASC, pointing to the importance of freshly isolated cells when investigating lipedema and emphasizing the impact of the milieu created by the surrounding adipose tissue cells in vivo.

Hence, we demonstrate the contribution of stromal vascular cell subpopulations, apart from mature adipocytes, to local tissue events involved in lipedema progression, including alterations within the vasculature. We were able to show that cells of the SVF are affected and impact endothelial barrier function in vitro without direct cell–cell interaction, although the factors responsible for the altered cellular function were not identified and will require further investigation. Despite a limited sample size, these results pose a huge impact on understanding the open questions in lipedema research.

## Figures and Tables

**Figure 1 biomedicines-10-01163-f001:**
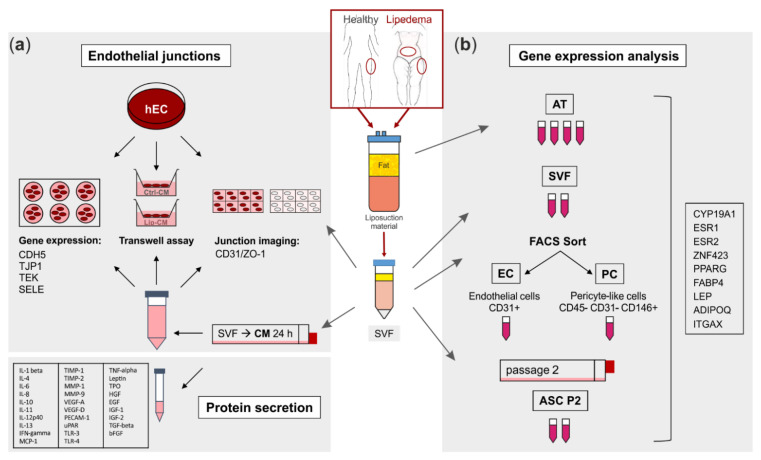
Schematic representation of the experimental set-up. Liposuction material of 4 healthy and 5 lipedema-affected subjects was used to isolate the stromal vascular fraction (SVF). (**a**) For fluorescence imaging of endothelial junctions, the SVF-contained endothelial cells (EC) of healthy and lipedema adipose tissue (AT) and human primary ECs (hEC) treated with conditioned media (CM) of healthy and lipedema SVF cells were stained for CD31 and ZO-1. After the treatment with CM, hECs were analyzed for endothelial permeability by a transwell assay and for gene expression of endothelial markers by quantitative real-time RT-PCR (qPCR). The CM of healthy individuals and lipedema patients were screened for protein secretion of inflammation and angiogenesis markers. (**b**) Samples of AT were collected from liposuction material of the thigh region of healthy and the thigh and abdominal regions of lipedema-affected subjects. CD31+ EC and CD45− CD31− CD146+ pericytes (PC) were isolated by fluorescence-activated cell sorting (FACS) of SVF. To achieve an adipose-derived stromal/stem cell (ASC) culture, SVF was cultivated for two passages. RNA was isolated from AT, SVF, EC, PC and ASC for gene expression analysis by qPCR.

**Figure 2 biomedicines-10-01163-f002:**
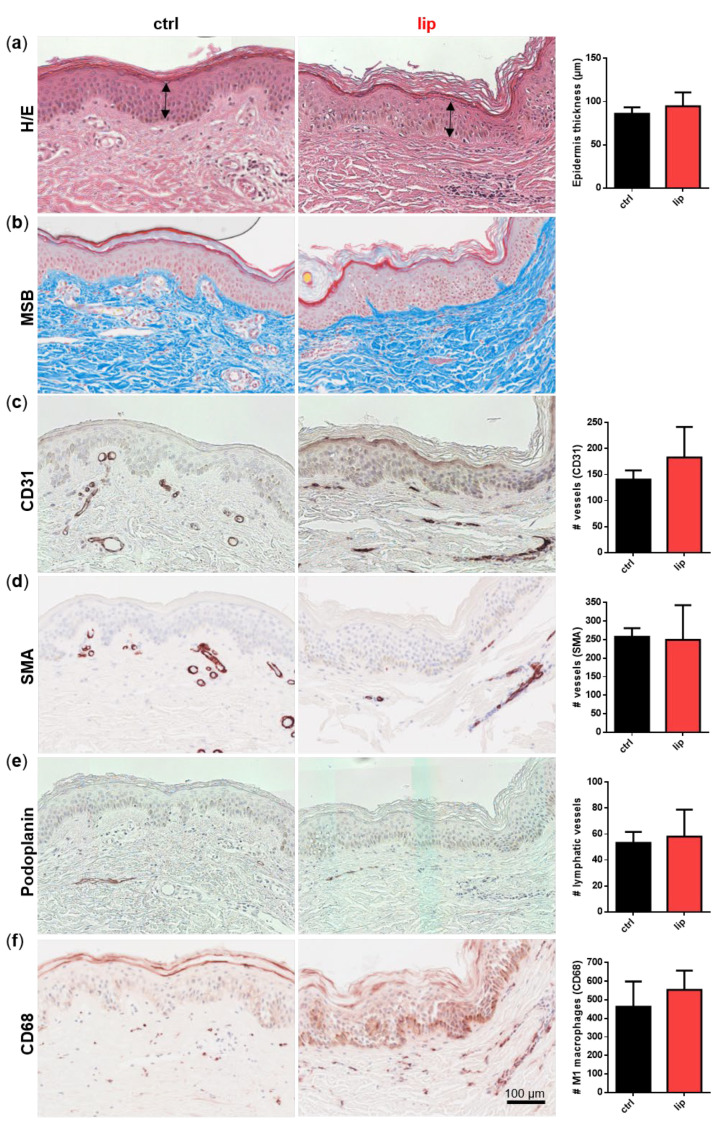
Histological analysis of consecutive sections of skin biopsies derived from healthy (ctrl: *n* = 3) and lipedema (lip: *n* = 3) individuals. For an overview, stainings were performed with hematoxylin and eosin (H/E) for detecting fibrin and collagens with Martius Scarlet blue (MSB), endothelial cells (EC) with CD31, pericytes with smooth muscle actin (SMA), lymphatic EC with podoplanin and macrophages with CD68. Epidermis thickness (H/E) and collagen fibers (MSB) showed no differences between ctrl and lip tissue (**a**,**b**). The number of CD31 positive vessels was slightly but not significantly increased (**c**), whereas high numbers of macrophages (CD68) could be observed in denser areas of capillaries in lipedema tissue (**f**), although this observation could not be confirmed when quantifying the whole tissue section. No differences could be observed in SMA+ vessels and lymphatic EC between ctrl and lip (**d**,**e**). Data are presented as mean ± SD. Statistical analysis: Student’s two-tailed *t*-test. Scale bar 100 µm.

**Figure 3 biomedicines-10-01163-f003:**
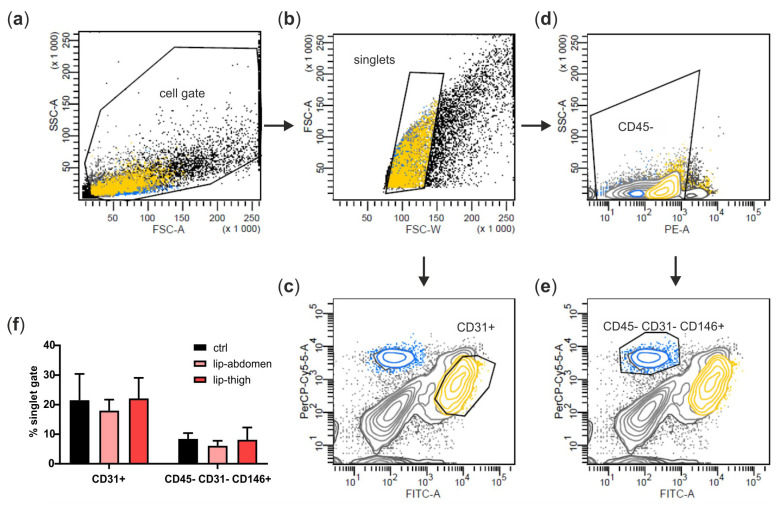
Isolation of perivascular cell populations from stromal vascular fraction (SVF) of abdominal and thigh adipose tissue. (**a**–**e**) Representative gating strategy for fluorescence-activated cell sorting (FACS) of the SVF. (**a**) SVF of healthy (ctrl) and lipedema (lip)-affected individuals were identified as cell gate on forward scatter (FSC) and side scatter (SSC), and (**b**) cell aggregates were excluded in an FSC-W versus FSC-A gate. (**c**) Singlets were further gated and sorted for CD31+ endothelial cells (EC) or (**d**) gated for CD45− cells, which were in turn gated and sorted for CD31− CD146+ pericytes (PC). (**f**) Data were analyzed using BD FACS Diva Software 5.03, and population frequencies were expressed as percent of the singlet cell gate. Data are presented as mean ± SD. Statistical analysis: Student’s two-tailed *t*-test. Abbreviations: Ctrl: healthy individuals thigh AT, lip-abdomen: lipedema abdominal AT, lip-thigh: lipedema thigh AT.

**Figure 4 biomedicines-10-01163-f004:**
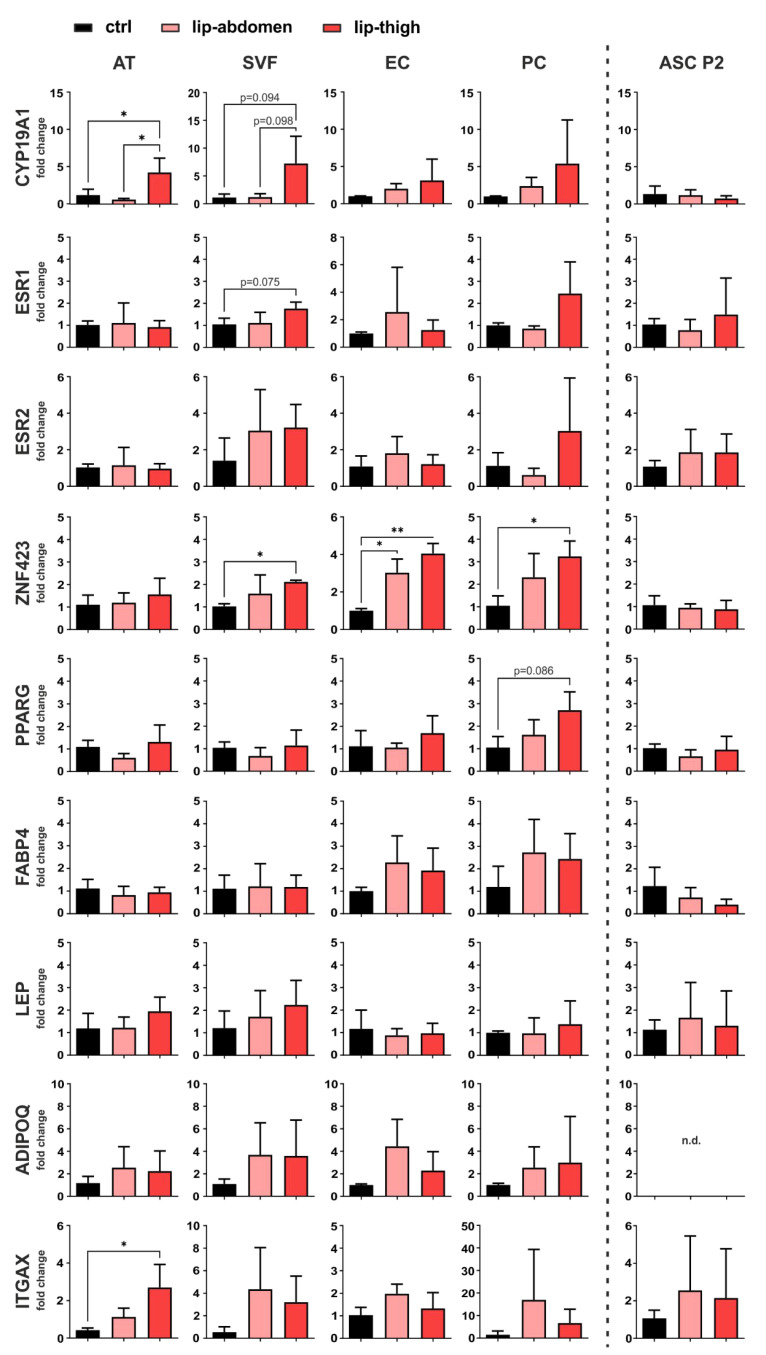
Gene expression analysis of factors involved in local estrogen metabolism, adipogenesis and inflammation. Subcutaneous adipose tissue (AT) of the thigh region collected from healthy individuals (ctrl: *n* = 3) and lipedema patients (lip-thigh: *n* = 4), as well as abdominal AT of lipedema patients (lip-abdomen: *n* = 3) served to isolate the stromal vascular fraction (SVF), which was in turn sorted for CD31+ endothelial cells (EC) and CD45− CD31− CD146+ pericytes (PC). SVF was cultivated and passaged two times to collect adipose-derived stromal/stem cells (ASC P2). RNA was isolated and quantitative real-time RT-PCR for gene expression analysis was performed. Fold change in gene expression (log scale) is normalized to ctrl expression levels and shown as mean ± SD. The *p*-values indicate the statistical significance of the group comparison (one-way ANOVA, Tukey’s post-hoc test); * *p* < 0.05, ** *p* < 0.01. Abbreviations: *CYP19A1*: cytochrome P450 family 19 subfamily A member 1, aromatase; *ESR1*: estrogen receptor 1; *ESR2*: estrogen receptor 2; *ZNF423*: zinc finger protein 423; *PPARG*: peroxisome proliferator-activated receptor gamma; *FABP4*: fatty acid binding protein 4; *LEP*: leptin; *ADIPOQ*: adiponectin; *ITGAX*: Integrin subunit alpha X, CD11C, n.d.: not detected.

**Figure 5 biomedicines-10-01163-f005:**
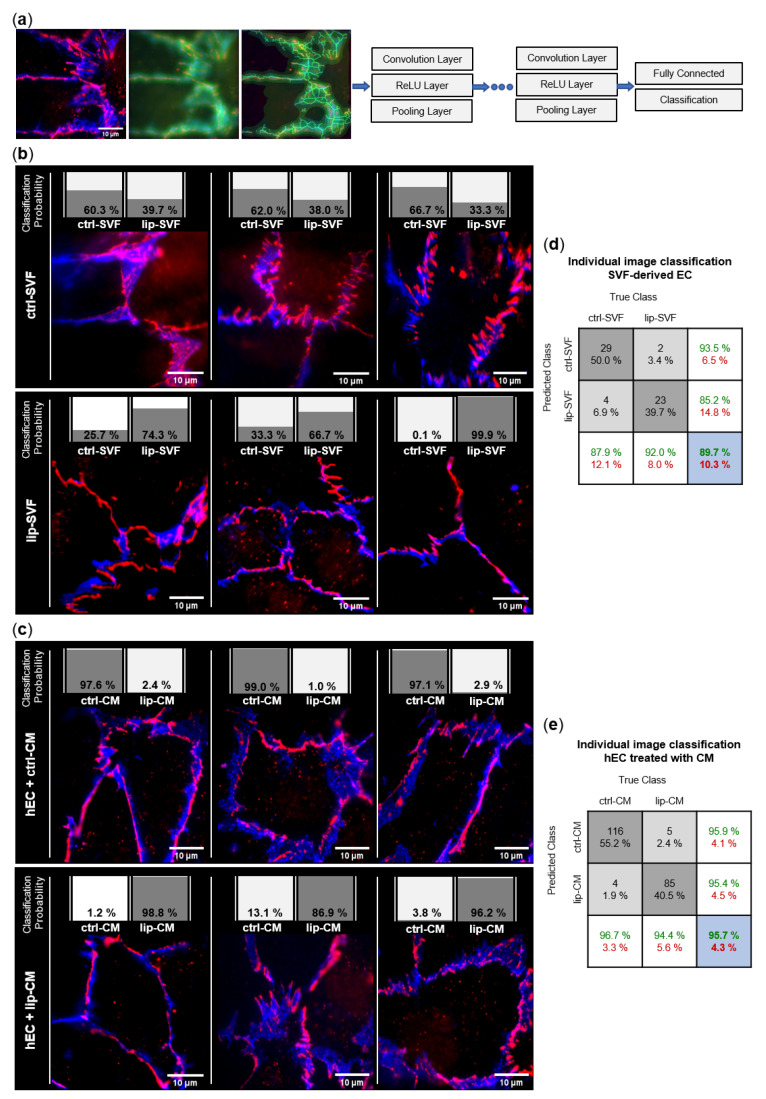
Convolutional neural network (CNN)-based analysis of endothelial junction morphology. Endothelial cell (EC) junctions present in the adipose tissue-derived stromal vascular fraction (SVF) and human primary ECs (hEC) treated with SVF conditioned media (CM) of healthy (ctrl: *n* = 3) and lipedema (lip: *n* = 3) subjects were labeled with anti-CD31-AlexaFluor488 (blue) and anti-ZO-1-AlexaFluor647 (red) antibodies. The deep learning method CNN was applied to predict the classification of individual images to the ctrl and lip groups. (**a**) Image augmentation and architecture of the CNN network used for the classification of the image data. Each grouped CNN hidden layer includes a convolution layer, a rectified linear unit (ReLU) layer and a pooling layer. The output layers form a fully connected layer and a classification layer. (**b**) Classification of 3 representative images of ctrl and lip SVF-derived EC and (**c**) ctrl and lip-CM treated hEC. The individual classification probability of the regarding image is shown as a bar chart. (**d**,**e**) Confusion matrices of individual immunofluorescence image classification. Columns show the true class of images used in the data set. Rows show the predicted class calculated by the CNN. Dark grey boxes: absolute and relative correct classification; light grey boxes: absolute and relative false classification; white boxes: cumulative percentage of correct (green) and false (red) classification; blue: average correct (green) and false (red) classification accuracy.

**Figure 6 biomedicines-10-01163-f006:**
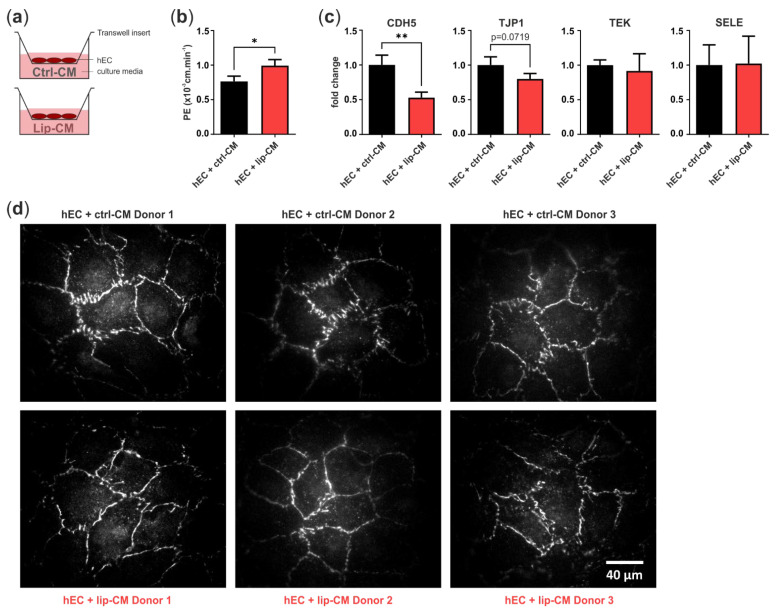
Transwell assay for endothelial permeability. (**a**) Schematic representation of the assay design. Human primary endothelial cells (hEC) were cultivated on porous transwell inserts and cultured with conditioned medium (CM) of stromal vascular fraction (SVF) cells isolated from thigh adipose tissue of healthy individuals (ctrl-CM, *n* = 3) and lipedema patients (lip-CM, *n* = 4). (**b**) Lip-CM treatment increased endothelial permeability, determined by calculation of the endothelial permeability coefficient (PE). (**c**) Quantitative real-time RT-PCR of endothelial junction markers revealed a reduced expression of *CDH5* and *TJP1* by hEC after treatment with lip-CM, whereas *TEK* and *SELE* were not affected. Fold change in gene expression (log scale) is normalized to ctrl expression levels. Data are presented as mean ± SD. Statistical analysis: Student’s two-tailed *t*-test; * *p* < 0.05, ** *p* < 0.01. (**d**) Representative fluorescence images of ZO-1-AF647-labeled hEC monolayer grown on transwell inserts and treated with either ctrl- or lip-CM (3 of 4 donors are shown). Abbreviations: *CDH5*: cadherin 5, VE-cadherin; *TJP1*: tight junction protein 1, ZO-1; *TEK*: TEK receptor tyrosine kinase, Tie-2; *SELE*: E-selectin.

**Table 1 biomedicines-10-01163-t001:** Semi-quantitative expression of 29 proteins involved in endothelial function and inflammation analyzed in conditioned media of ctrl (*n* = 3) and lip (*n* = 3) SVF cells.

Protein Name	ctrl (A.U. Mean ± SD)			lip (A.U. Mean ± SD)			*p*-Value
IL-8	1004	±	639	191	±	100	0.049
TIMP-1	404	±	204	856	±	455	0.176
TIMP-2	287	±	164	525	±	309	0.287
IL-12p40	415	±	364	631	±	243	0.386
IL-11	336	±	135	414	±	95	0.408
IL-6	621	±	99	4149	±	6814	0.422
MCP-1	2717	±	348	2998	±	702	0.558
bFGF	249	±	190	319	±	168	0.627
IL-13	105	±	154	63	±	75	0.648
VEGF-D	169	±	127	217	±	148	0.671
IGF-1	350	±	310	277	±	132	0.684
MMP-1	5711	±	3794	7004	±	4061	0.686
uPAR	285	±	273	357	±	198	0.700
IGF-2	185	±	192	231	±	170	0.750
TLR3	47	±	60	64	±	73	0.755
HGF	220	±	214	167	±	216	0.757
PECAM-1	165	±	170	134	±	123	0.791
MMP-9	203	±	178	235	±	146	0.803
TGF-beta	264	±	321	322	±	309	0.821
VEGF-A	197	±	137	227	±	181	0.825
IFN-gamma	276	±	231	306	±	133	0.834
IL-10	127	±	130	143	±	106	0.860
TLR4	90	±	89	81	±	61	0.880
TPO	116	±	129	104	±	105	0.899
IL-4	220	±	206	238	±	168	0.902
IL-1beta	197	±	163	206	±	136	0.936
Leptin	68	±	83	66	±	72	0.972
EGF	8696	±	3611	8709	±	2820	0.996
TNF-alpha	186	±	169	186	±	156	0.999

Statistical analysis: Student’s two-tailed *t*-test. Abbreviations: Ctrl: control; lip: lipedema; A.U.: arbitrary units; IL: interleukin; IFN: interferon; MCP: monocyte-chemoattractant protein; TIMP: tissue inhibitor of metalloproteinases; MMP: matrix metallopeptidase; TNF: tumor necrosis factor; uPAR: plasminogen activator surface receptor; TPO: thyroid peroxidase; PECAM: platelet and endothelial cell adhesion molecule; HGF: hepatocyte growth factor; EGF: epidermal growth factor; IGF: insulin-like growth factor; TGF: transforming growth factor; bFGF: basic fibroblast growth factor; VEGF: vascular endothelial growth factor; TLR: toll-like receptor.

## Data Availability

The data presented in this study are available from the corresponding author, upon reasonable request.

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
