# Peer review of "Multi-Level Analysis of Adipose Tissue Reveals the Relevance of Perivascular Subpopulations and an Increased Endothelial Permeability in Early-Stage Lipedema"

_biomedicines, 2022, doi:10.3390/biomedicines10051163_

Round 1

Reviewer 1 Report

In this study the authors utilize histological, immunofluorescence imaging, gene and protein expression to identify cell types and molecular changes that contribute to the onset and progression of lipedema. While the n for this study is fairly low, the methods and experimental design lead to some novel characterization of factors that contribute to lipedema, particularly the weakened endothelial barrier. I’m not sure the currently targeted special issue is the most appropriate for this article. Seems like this would be a better fit for “10th Anniversary of Biomedicines—Advances in Development of Human Adipose Cells” (Biomedicines | Special Issue : 10th Anniversary of Biomedicines—Advances in Development of Human Adipose Cells (mdpi.com)

  1. I would consider creating a new paragraph at line 59 at the sentence that starts with different.
  2. Can you clarify the n for samples used for histological analysis. I assume 4 based on the n provided for number of participants sampled, however as a reader I always find it helpful to have the n provide in the figure caption for reference. If you can do this in the figure captions throughout that would be optimal. It’s present in some but not all.
  3. Is there an explanation somewhere that clarifies why ctrl n = 4 and lip n = 5 described at lines 102-103 but then in some of the figures ctrl n = 3 and lip n = 4? I may have just missed it, but if this is not described in the text please include a justification for this discrepancy. 
  4. You have a lot of very dense figures in this manuscript. Ultimately, there were no significant differences found in figure 2. I would recommend reporting the quantitative endpoints and corresponding statistics in a table and moving all of the histology images to supplemental.
  5. It would be helpful to letter each panel in figure 3 so that when you reference FSC, SSC, etc in the figure caption it is easier for the reader to reference which plot they should be looking at.
  6. In the results section, F (between df, within df) and U values should be reported with p-values where applicable.
  7. Figure 4 is very overwhelming and I think makes it hard to see what the take home message is/what the significant findings really are. I think this would be better represented by only displaying graphs for significant findings and reporting the rest of the genes assessed and their findings in a table. Histograms for findings that were not significant could always be included in supplemental. But here all of those genes that showed no significant changes are distracting from those that were significant.
  8. Line 443, I’m not convinced by the reported data that you can say involvement of estrogen signaling, metabolism yes, but changes in gene expression does not support changes in signaling. This could always be a discussion point (i.e. local changes in estrogen metabolism are likely to impact estrogen signaling in this tissue although that was not observed here), but the current way this is phrased in the results section insinuates you found changes in receptors.
  9. Figure 7, similar suggestions as Figure 4. Lose all the figures with data that is not significant, which is everything but IL8 and consolidate the rest into a table and/or move the remaining figures to supplemental.

Author Response

Response to Reviewer 1 Comments

Point 1: I would consider creating a new paragraph at line 59 at the sentence that starts with different.

Response 1: Thank you for reviewing our manuscript and your valuable comments. We adapted the text accordingly.

Point 2: Can you clarify the n for samples used for histological analysis. I assume 4 based on the n provided for number of participants sampled, however as a reader I always find it helpful to have the n provide in the figure caption for reference. If you can do this in the figure captions throughout that would be optimal. It’s present in some but not all.

Response 2: We added the information to the figure legends.

Point 3: Is there an explanation somewhere that clarifies why ctrl n = 4 and lip n = 5 described at lines 102-103 but then in some of the figures ctrl n = 3 and lip n = 4? I may have just missed it, but if this is not described in the text please include a justification for this discrepancy. 

Response 3: During the course of the project new assays were added. Since the experiments were conducted with freshly isolated cell material, we could not repeat the new assays with previous donors. Therefore, the total of 4 healthy and 5 lipedema donors could not be reached in every assay.

Point 4: You have a lot of very dense figures in this manuscript. Ultimately, there were no significant differences found in figure 2. I would recommend reporting the quantitative endpoints and corresponding statistics in a table and moving all of the histology images to supplemental.

Response 4: In lipedema research one major method of choice for differential diagnosis is histology of the affected adipose tissue with the typical morphology, hypertrophy, macrophage infiltration etc., in lipedema (Al-Ghadban et al. DOI: 10.1155/2019/8747461, Felmerer et al. DOI: 10.1016/j.jss.2020.03.055. and DOI: 10.1038/s41598-020-67987-3). According to our findings, early-stage lipedema, in contrast, cannot be differentiated from healthy individuals in histology. In consequence, the lack of significant differences to healthy tissue is actually the relevant aspect of this figure. Therefore, we think this information should be displayed as main figure.

Point 5: It would be helpful to letter each panel in figure 3 so that when you reference FSC, SSC, etc in the figure caption it is easier for the reader to reference which plot they should be looking at.

Response 5: We adapted the figure accordingly.

Point 6: In the results section, F (between df, within df) and U values should be reported with p-values where applicable.

Response 6: We are very thankful for your revision. Because of your comment, we found and error in the methods. For the qPCR statistics, we used One-way ANOVA and multiple comparison Tukey post-hoc test and not the Dunnet’s test as described, we corrected this error accordingly. For easier reading we added the statistical tests used in this study to each single figure caption. Since we have a huge amount of data, adding the suggested values in addition to our p-values into the main text would make it difficult to read. For detailed insight, we are of course open to share our raw data with everybody upon request, as stated in the Data Availability Statement.

Point 7: Figure 4 is very overwhelming and I think makes it hard to see what the take home message is/what the significant findings really are. I think this would be better represented by only displaying graphs for significant findings and reporting the rest of the genes assessed and their findings in a table. Histograms for findings that were not significant could always be included in supplemental. But here all of those genes that showed no significant changes are distracting from those that were significant.

Response 7: Thank you for this suggestion. We agree that Figure 4 is an extensive representation of our qPCR data. Because the finding that respective genes were found unaffected by early-stage lipedema in our study is contradicting reports of other groups in the field, we think that all of the data - also that not significantly different from healthy- is relevant for the field and should be prominently displayed. We tried to transform the data into a table as suggested, but it was very difficult to interpret. For a better overview and to minimize the figure we moved the respective dCT values from Figure 4b to the supplements (new supplementary figure 1).

Point 8: Line 443, I’m not convinced by the reported data that you can say involvement of estrogen signaling, metabolism yes, but changes in gene expression does not support changes in signaling. This could always be a discussion point (i.e. local changes in estrogen metabolism are likely to impact estrogen signaling in this tissue although that was not observed here), but the current way this is phrased in the results section insinuates you found changes in receptors.

Response 8: We agree that changes in signaling cannot be concluded from analyzing solely qPCR and altered the respective sentence accordingly.

Point 9: Figure 7, similar suggestions as Figure 4. Lose all the figures with data that is not significant, which is everything but IL8 and consolidate the rest into a table and/or move the remaining figures to supplemental.

Response 9: Thank you for this suggestion. We transformed figure 7 into a table.

Reviewer 2 Report

This is a very good study in which the authors examined the different cell types involved in the pathogenesis of lipedema, using fat and stromal vascular fraction   isolated from subcutaneous adipose tissue of lipidema-affected patients. The results are interesting  and suggest that lipedema is associated with a weakened endothelial barrier and that the effect of estrogen is mediated through increased expression of aromatase in preadipocytes and ZNF-423 in perivascular cell subpopulations.

The experiments were performed thoroughly, without major methodological shortcomings

The manuscript is very well written, the results are well documented and recent findings are carefully discussed.

I have only minor comment:

Line 685:The  acronym “EV” appears for the first time and should be defined.

The authors analysed adipogenesis according to ZNF423, PPARγ, FABP4, leptin and adiponectin gene expression. Only ZNF423 and to a lesser extent PPARγ were upregulated in lipedema patients compared to control group. In order to develop adipocytes from precursor cells, the mechanisms involved in utilization of lipid storage and lipolysis must be activated. Do the authors explain why the expression of more specific genes involved in glucose utilization, triacylglycerol synthesis, regulation of the release of non-esterified fatty acids in adipocytes have not been tested.

Author Response

Response to Reviewer 2 Comments

Point 1: The  acronym “EV” appears for the first time and should be defined.

Response 1: Thank you for reviewing our manuscript and your valuable comments. The abbreviation “EV” is defined in line 82.

Point 2: The authors analysed adipogenesis according to ZNF423, PPARγ, FABP4, leptin and adiponectin gene expression. Only ZNF423 and to a lesser extent PPARγ were upregulated in lipedema patients compared to control group. In order to develop adipocytes from precursor cells, the mechanisms involved in utilization of lipid storage and lipolysis must be activated. Do the authors explain why the expression of more specific genes involved in glucose utilization, triacylglycerol synthesis, regulation of the release of non-esterified fatty acids in adipocytes have not been tested.

Response 2: It would indeed be highly interesting to study specifically the adipocyte population or alternatively the potential of perivascular cells to differentiate to adipocytes, including genes of the lipid metabolism, citrate cycle or for example the status of mitochondria, this was however not the scope of the current study.

Here, we have focused exclusively on the stromal/perivascular and vascular population of cells within the adipose tissue and excluded mature adipocytes from our analyses, which where only considered in the adipose tissue samples in the qPCR analysis. Within this study, we did not characterize in vitro adipogenesis, which is typically done with ASC (this was subject of a previous study by our group). Here, we characterized the current status of freshly isolated perivascular and stromal cells. The cellular fraction of adipocytes with lipid accumulation, is already excluded during isolation by centrifugation and is therefore not reflected in the SVF population. In consequence, our analyses did not focus on the metabolism of adipocytes. We were interested on the one hand in the early commitment of perivascular cells to the adipocyte lineage (ZNF423, PPARG) and on the other hand in adipokine expression of the adipose tissue (adiponectin, leptin), due to their invovlment in angiogenesis.